 RESEARCH ADVANCE

# Humans adapt rationally to approximate estimates of uncertainty

Erdem Pulcu[1]*, Michael Browning[1,2]

[1]Department of Psychiatry, University of Oxford, Oxford, United Kingdom; [2]Oxford Health NHS Trust, Oxford, United Kingdom

## eLife Assessment

This study makes an **important** contribution by showing that humans adapt learning rates rationally to environmental volatility yet systematically misattribute noise as volatility, demonstrating approximate rationality with simplified internal models. The evidence is **compelling**, encompassing a cleverly designed volatility-versus-noise paradigm, innovative lesion-based comparisons between reinforcement-learning and degraded Bayesian Observer Models, and convergent behavioural and pupillometric data. Expanding formal model comparisons (e.g., BIC/AIC) and directly contrasting RL and Bayesian fits to physiological markers would further enhance the work, but these are minor limitations that do not detract from the core findings.

*For correspondence:
pulerd@gmail.com

**Abstract** Efficient learning requires estimation of, and adaptation to, different forms of uncertainty. If uncertainty is caused by randomness in outcomes (noise), observed events should have less influence on beliefs, whereas if uncertainty is caused by a change in the process being estimated (volatility) the influence of events should increase. Previously, we showed that humans respond appropriately to changes in volatility irrespective of outcome valence (Pulcu and Browning, 2017), but there is less evidence of a rational response to noise. Here, we test adaptation to variable levels of volatility and noise in human participants, using choice behaviour and pupillometry as a measure of the central arousal system. We find that participants adapt as expected to changes in volatility, but not to changes in noise. Using a Bayesian observer model, we demonstrate that participants are, in fact, adapting to estimated noise, but that their estimates are imprecise, leading them to misattribute it as volatility and thus to respond inappropriately.

## Introduction

It is much easier to respond appropriately to an event if we know what has caused it. For example, if heavy traffic means that our drive into work takes longer than normal, the best course of action the next time we have to make the journey depends on what caused the traffic to be heavier (*Yu and Dayan, 2005*). If it was caused by a one-off or random event, such as a broken-down lorry, then we should continue using the same route as before, whereas if it was caused by some longer-term change, perhaps there are new road works nearby disrupting the traffic, we should consider a different route. Frequently, however, the causes of events are not obvious. We experience the heavy traffic but aren't sure why it has occurred. In these situations, the best we can do is make an educated guess, based on our experience, about what broad type of causal process has led to recent events. In the case of the drive into work, if the traffic has been heavier for a number of days in a row it is likely that some prolonged shift has occurred, and we should change routes, whereas if the traffic changes noisily from day to day, then we should probably stick with our usual route. In the learning literature, this problem is often framed as a competitive attribution of uncertainty

to one of two types; expected uncertainty, which is caused by the variability of noisy associations, and unexpected uncertainty, which is caused by longer-lasting changes (sometimes called volatility) in an association (*Behrens et al., 2007*; *Browning et al., 2015*; *Nassar et al., 2012*; *Pulcu and Browning, 2017*; *Yu and Dayan, 2005*). The behavioural importance of this attribution process can be seen in the driving example given above; an event caused by noise requires the opposite behavioural response (continuing to use the same route) than the same event caused by volatility (switching routes). Consequently, effective decision making often depends on the accurate attribution of uncertainty, with misattribution having a substantial detrimental effect on choice (*Pulcu and Browning, 2019*).

The influence of events on subsequent choice can be estimated within a reinforcement learning framework as the learning rate parameter (*Sutton and Barto, 2018*), with a higher learning rate indicating a greater influence of the event on behaviour. As described above, the normative response to changes in volatility and noise is to use a higher learning rate when volatility is high and/or noise is low (*Pulcu and Browning, 2019*; *Yu and Dayan, 2005*). A large number of studies have found the predicted increase in learning rates in response to higher outcome volatility in human learners (*Behrens et al., 2007*; *Behrens et al., 2008*; *Browning et al., 2015*; *Gagne et al., 2020*; *Nassar et al., 2012*; *Pulcu and Browning, 2017*). In contrast, the evidence for adaptation of learning in response to changes in outcome noise is less complete. Previous studies have described the expected reduction of learning rates when outcome noise is high, but only when the level of noise is explicitly signalled in a task (*Diederen and Schultz, 2015*), or when it is made unambiguous by virtue of being very much smaller than changes in outcome caused by volatility (*Nassar et al., 2010*; *Nassar et al., 2012*). As illustrated in the driving example above, we are often faced with situations in which there exists significant uncertainty about whether an event has been caused by volatility or noise. To date, however, the degree to which human learners are able to discriminate between these types of uncertainty, when they are not explicitly labelled, has not been closely examined.

From a neurobiological perspective, activity of central modulatory neurotransmitter systems have been argued to represent distinct sources of uncertainty during learning, with central norepinepheric (NE)/locus coeruleus (LC) activity described as signalling changes in the associations (i.e. volatility) and central cholinergic activity representing noise (*Yu and Dayan, 2005*). Electrophysiological measures of LC activity in non-human primates have been shown to correlate with pupil dilation (*Joshi et al., 2016*), suggesting it may be possible to estimate activity in this system in humans using pupillometry. Taking this approach, indirect support for this role of the NE system has been provided by studies of human participants that report greater pupillary size in volatile relative to stable contexts (*Browning et al., 2015*; *Nassar et al., 2012*; *Pulcu and Browning, 2017*). However, the pupil also responds to other learning signals, such as surprise (*Browning et al., 2015*; *O'Reilly et al., 2013*; *Preuschoff et al., 2011*) and has been reported as being smaller when outcome noise is high (*Nassar et al., 2012*). Neuroimaging evidence suggests an association between activity in other central neurotransmitter nuclei, including the cholinergic basal forebrain, and pupil dilation (*de Gee et al., 2017*). Overall, this suggests that the pupillary signal may reflect a more general belief updating process (*O'Reilly et al., 2013*) rather than a specific volatility signal and thus that, like learning rates, pupil size should increase when noise is reduced as well as when volatility is increased.

In this paper, we test whether human participants modify their learning in situations in which the attribution of uncertainty as volatility or noise is challenging (*Figure 1a–c*). We report the results of a study in which participants completed a learning task during which the noise and volatility of both win and loss outcomes were independently manipulated. Participant behaviour was characterised using learning rate parameters derived from reinforcement learning models of choice, while interpretation of the results was facilitated by a Bayesian Ideal Observer model that was developed to provide a benchmark comparator to participant behaviour (*Behrens et al., 2007*; *Nassar et al., 2012*; *Piray and Daw, 2021*; *Pulcu et al., 2022*) and by the collection of pupillometry data as a physiological marker of central neurotransmitter function (*de Gee et al., 2017*; *Joshi et al., 2016*). It was predicted that human participants would be able to adapt appropriately to the cause of the events they encountered—using a higher learning rate, and displaying increased pupil size, when volatility was high and when noise was low for both win and loss outcomes (*Figure 1d*).

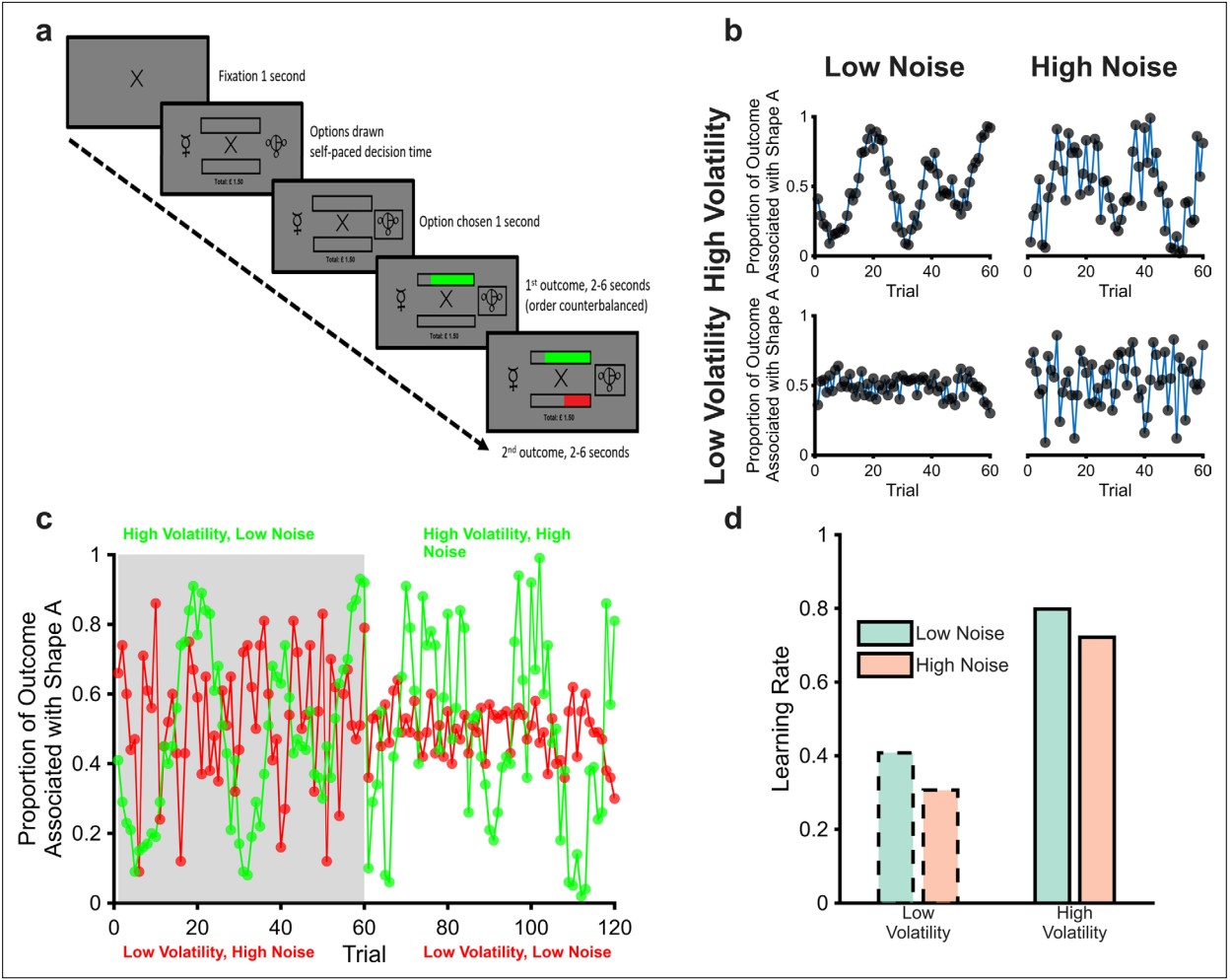

**Figure 1.** The magnitude learning task (**a**) Timeline of one trial from the learning task. On each trial, participants were presented with two abstract shapes and were asked to choose one of them. The empty bars above and below the fixation cross represented the total available wins and losses for the trial, the full length of each bar was equivalent to £1. Participants chose a shape and then were shown the proportion of each outcome that was associated with their chosen shape as coloured regions of the bars (green for wins and red for losses). The empty portions of the bars indicated the win and loss magnitudes associated with the unchosen option, allowing participants to infer which shape would have been the better option on every trial. The task consisted of six blocks of sixty trials each. The volatility and noise of the two outcomes varied independently between blocks with different shapes used in each block. Panel (**b**) illustrates outcomes from the four block types. As can be seen, blocks with high volatility and low noise (top left), and those with low volatility and high noise (bottom right), present participants with a similar range of magnitudes. Participants, therefore, have to distinguish whether variability in the outcomes is caused by volatility or noise from the temporal structure of the outcomes rather than the size of changes in magnitude (cf. *Diederen and Schultz, 2015*; *Krishnamurthy et al., 2017*; *Nassar et al., 2012*). Panel (**c**) shows two example blocks (one block in grey, the other in white) with both win (green) and loss outcomes (red) displayed. Panel (**d**) shows the expected adaptation of learning rates in response to the manipulation of volatility and noise; for both win and loss outcomes, learning rates should be increased when volatility is high and when noise is low.

## Results

### Participant demographics

70 participants (see *Table 1* for demographic information) completed a learning task in which they had to choose one of two stimuli based on the separately estimated magnitudes of win and loss outcomes associated with the stimuli (*Figure 1*). Participants were able to learn the best option to choose in the task, selecting the most highly rewarded option on an average of 71% of trials (range 65–74%).

**Table 1.** Demographic details of participants.

| Measure | Mean (SD) |
| --- | --- |
| Age | 29.07 (10.86) |
| Gender | 69% Female |
| QIDS-16 | 5.26 (4.25) |
| Trait-STAI | 36.21 (10.42) |
| State-STAI | 30.29 (8.57) |

QIDS-16; Quick Inventory of Depressive Symptoms, 16-item self-report version. Trait/State-STAI; Spielberger State-Trait Anxiety Inventory.

## Experimental manipulation of volatility and noise influences participant choice behaviour

As explained above, high levels of volatility and low levels of noise should increase the degree to which outcomes influence choice behaviour. A crude metric of this effect is provided by examining participant choice as a function of the previous outcome. In the task, a win outcome of >50 p or a loss outcome of <50 p associated with Shape A prompts participants to select Shape A in the subsequent trial, with the other outcomes (i.e. win <50 p and loss >50 p) prompting choice of Shape B. The influence of the outcomes on choice can, therefore, be roughly estimated as the relative proportion of trials in which Shape A was chosen when it was prompted by a previous outcome of a given magnitude, compared to when Shape B was prompted. Analysis of this choice metric (*Figure 2a–b*) found the expected effect of volatility, with participant choice being more influenced by previous outcomes when volatility was higher ($F_{(1,696)}=99.8$, p<0.001). An effect of noise was observed, but in the opposite direction to expected, with outcomes influencing choice more when noise was increased ($F_{(1,696)}=4.79$, p=0.03). No significant difference between the influence of win and loss outcomes was found ($F_{(1,696)}=1$, p=0.32), and there was no interaction between volatility and noise ($F_{(1,693)}=0.61$, p=0.4). Having found some evidence of an impact of the uncertainty manipulations on a crude measure of subject choice, we next sought to characterise this effect using reinforcement learning models fitted to participant choices.

## Participants adjust normatively to changes in volatility but not noise

We aimed to capture the computational process that underlies participant choice behaviour by fitting different reinforcement learning models to choice data separately for each block of the task and each participant. The best fitting RL model included separate learning rates for win and loss outcomes allowing estimation of the degree to which participants adjusted these learning rates in response to the block-wise changes in outcome volatility and noise (see Supplementary materials and methods for model comparison and selection analyses).

Consistent with the analysis of choice data reported above, there was a significant main effect of volatility (*Figure 2c–d*; $F_{(1,696)}=22.2$, p<0.001), with a higher learning rate used when volatility was high. There was no main effect of noise ($F_{(1,696)}=0.63$, p=0.43) on learning rate or outcome valence ($F_{(1,696)}=0.15$, p=0.7). An interaction between volatility and noise ($F_{(1,693)}=7.74$, p=0.006) was also significant. A higher volatility led to a significantly raised learning rate when noise was low ($F_{(1,383)}=27.1$, p<0.01), with a non-significant increase when noise was high ($F_{(1,311)}=1.13$, p=0.29). Higher noise was associated with a non-significant reduction in learning rates when volatility was high ($F_{(1,347)}=2.57$, p=0.11) but to a significant increase in learning rate when volatility was low ($F_{(1,347)}=4.7$, p=0.031).

In summary, analysis of both crude choice data and learning rates indicates that participants adapted appropriately to changes in the volatility of learned associations but did not show a consistent response to changes in noise. In the next section,, we utilise a Bayesian observer model (BOM) to investigate potential causes for this relative insensitivity to noise.

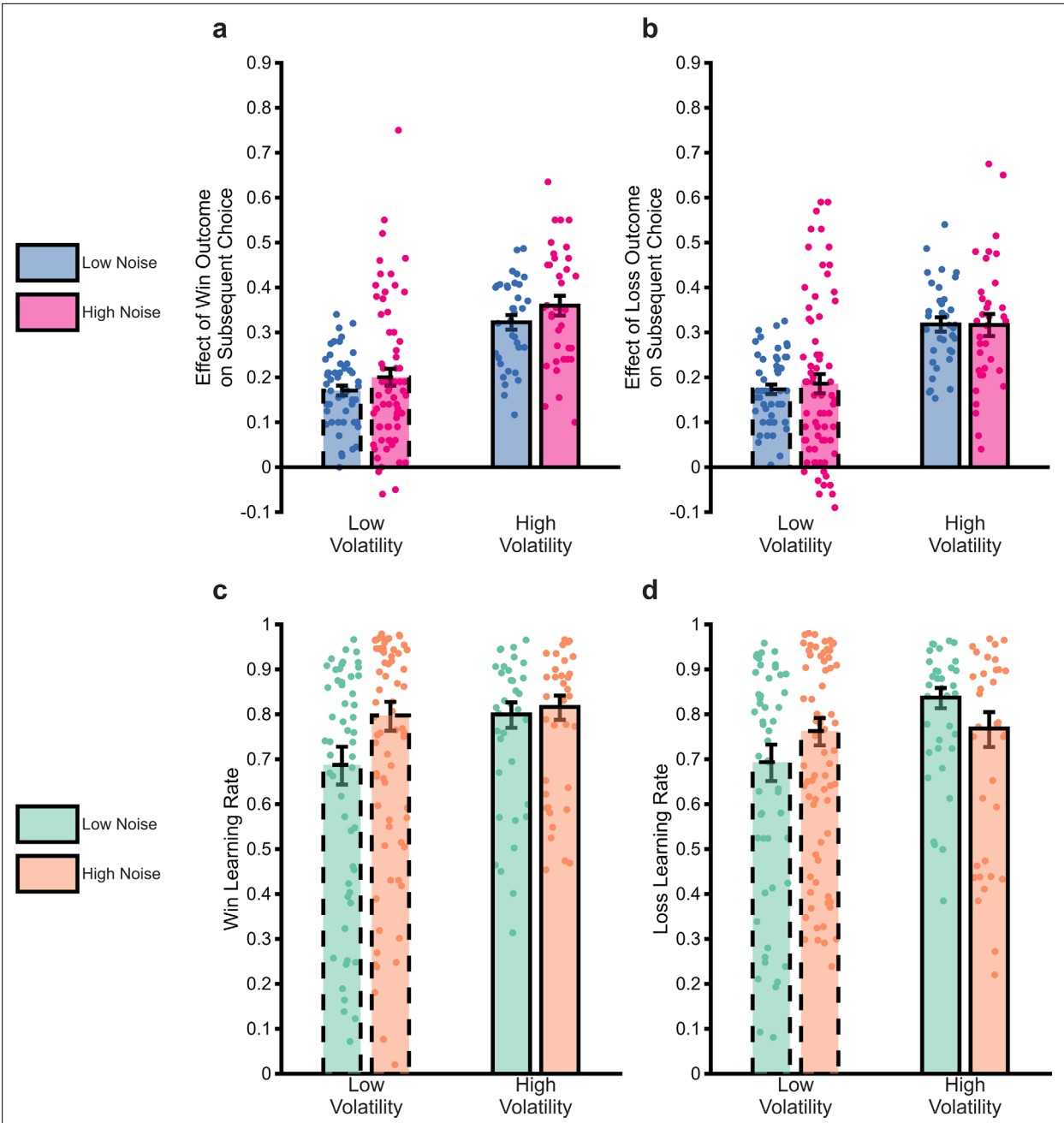

**Figure 2.** The impact of uncertainty manipulations on participant choice. Panels a and b report a summary metric for the effect of win and loss outcomes on subsequent choice. The metric was calculated as the proportion of trials in which an outcome of magnitude 51–65 associated with Shape A was followed by choice of the shape prompted by the outcome (i.e. Shape A for win outcomes, Shape B for loss outcomes) relative to when the outcome magnitude was 49–35 (see Methods and materials for more details). We focused on this outcome range as these range of magnitudes were covered by all volatility × noise conditions and it was dictated by the relatively smaller range coverage in the low volatility low noise condition (also see *Figure 1C*, loss outcomes shown in red between trials 60–120). The higher this number, the greater the tendency for a participant to choose the shape prompted by an outcome. As can be seen, the outcome of previous trials had a greater influence on participant choice when volatility was high, with a small effect of noise, in the opposite direction to that predicted. Panels c and d report the win and loss learning rates estimated from the same data. Again, the expected effect of volatility is observed, this time with no consistent effect of noise. Bars represent the mean (± SEM) of the data, with individual data points superimposed.

The online version of this article includes the following figure supplement(s) for figure 2:

**Figure supplement 1.** Comparison of estimated win and loss learning rates from the two estimation reinforcement learning (RL) models.

**Figure supplement 2.** Results of the generate-recover procedure for the reinforcement learning (RL) measurement model.

**Figure supplement 3.** Behaviour of a simple reinforcement learning (RL) model fit across all task blocks.

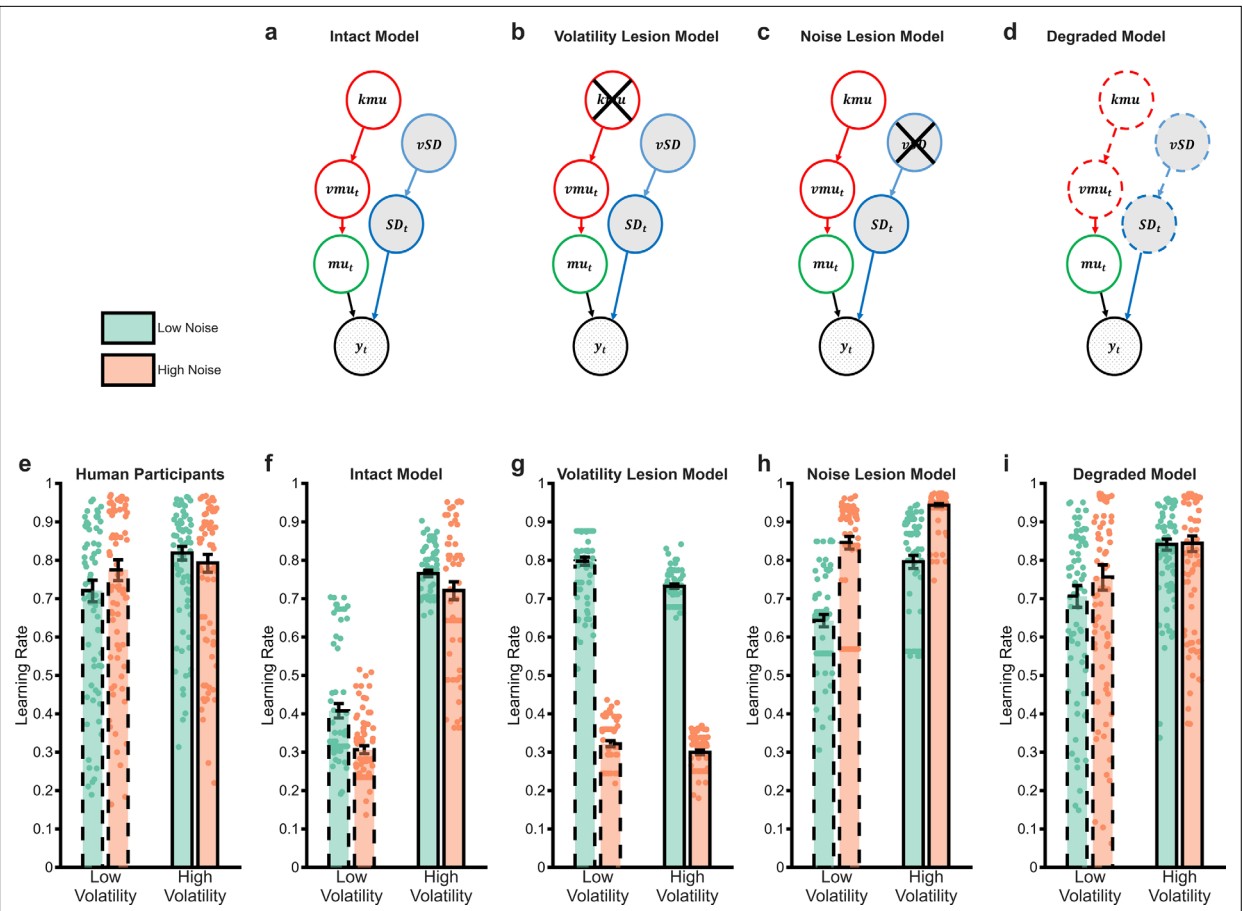

**Figure 3.** The behaviour of Bayesian observer models. Bayesian Observer Models (BOM) invert generative descriptions of a process, indicating how an idealised observer may learn. We developed a BOM based on the generative model of the task we used (**a**) Details of the BOM are provided in the methods, briefly, it assumes that observations ($y_i$) are generated from a Gaussian distribution with a mean ($mu_i$) and standard deviation ($SD_i$). Between observations, the mean changes with the rate of change controlled by the volatility parameter ($vmu_i$). The standard deviation and volatility of this model estimate the noise and volatility described for the task. The last parameters control the change in volatility ($kmu$) and standard deviation ($vs$) between observations, allowing the model to account for different periods when these types of uncertainty are high and others when they are low. The BOM adjusts its learning rate in a normative fashion (**f**), increasing it when volatility is higher, or noise is lower. The BOM was lesioned in a number of different ways in an attempt to recapitulate the learning rate adaptation observed in participants (shown in panel **e**). Removing the ability of the BOM to adapt to changes in volatility (**b**) or noise (**c**) did not achieve this goal (**g, h**). However, degrading the BOMs representation of uncertainty (**d**) was able to recapitulate the behavioural pattern of participants (**i**) Bars represent the mean (± SEM) of participant learning rates, with raw data points presented as circles behind each bar.

The online version of this article includes the following figure supplement(s) for figure 3:

**Figure supplement 1.** Analysis of the behaviour of the latent-state model.

## Using a Bayesian observer model to characterise noise insensitivity

Bayesian observer models (BOM) can be used as normative benchmarks against which human behaviour may be compared (*Behrens et al., 2007*; *Nassar et al., 2012*; *Piray and Daw, 2021*; *Pulcu et al., 2022*). BOMs are generally not fit to participant choice, rather these models invert a generative process assumed to underlie observed events and provide an estimate of the belief of an idealised agent exposed to the same outcomes as participants. We developed a BOM (*Pulcu et al., 2022*) based on the generative process underlying the outcome magnitudes of our task (*Figure 3a*). The BOM explicitly estimates the volatility and noise of the outcomes and uses these estimates to influence its belief about the likely magnitude of upcoming outcomes (see methods for more details). We first tested whether the BOM reproduced the normative learning rate adaptation to changes in volatility and noise described in the introduction, by exposing the model to the same outcomes as participants and using the model's belief about the likely magnitude of the win and loss outcome on

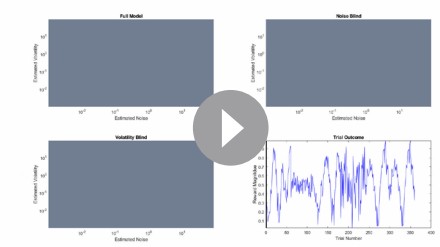

**Animation 1.** Estimation of the causes of uncertainty by the Bayesian Observer Model. Lower right panel: Synthetic data with periods of high (trial number: 0-60; 120-180; 240-360) and low (trial number: 60-120; 180-240) volatility and high (trial number: 180-240) and low (trial number: 1-180; 240-360) noise was provided to the Bayesian Observer Models. The models trial-by-trial estimate of volatility and noise is illustrated by marginalising over all but the *vmu* and *SD* dimensions of the joint probability distribution (see description of the Bayesian Observer Model in the methods). This produces a two-dimensional probability density of the model's estimate of volatility (y-axes) and noise (x-axes). The current data being fed to the model is illustrated by the solid line moving through the data. Top left panel: The estimated uncertainty of the full (unlesioned) model. The model adapts to different periods of high and low volatility reasonably well (e.g. see the period around trial 180 when the data moves from high volatility/low noise to high noise/low volatility). The fully lesioned models are provided for comparison (see methods section for a description of these models). Top right panel: The noise blind model (*vSD* has been removed) cannot account for changes in noise and so any change in either volatility or noise is captured as a change in volatility. Lower left panel: The volatility blind model (*kmu* has been removed) cannot account for changes in volatility and so any change in either volatility or noise is captured as a change in noise.

each trial to generate choices. We then estimated the effective learning rate of the model by fitting the same RL model used to analyse participants' choices to the model's choices. These learning rates are presented in *Figure 3f* (*Figure 3e* reproduces the learning rates of participants, averaged across wins and losses, for comparison). As can be seen, the BOM adapts as expected, using a higher learning rate both when volatility increases ($F_{(1,696)}=422$, $p<0.001$) and when noise decreases ($F_{(1,696)}=21.2$, $p<0.001$). No effect of outcome valence or interaction between volatility and noise (all $p>0.09$) was observed.

Having shown that an optimal learner adjusts its learning rate to changes in volatility and noise as expected, we next sought to understand the relative noise insensitivity of participants. In these analyses, we 'lesion' the BOM, to reduce its performance in some way, and then assess whether doing so recapitulates the pattern of learning rate adaptation observed for participants (*Figure 3e*). In other words, we damage the model so it performs less well and then assess whether this damage makes the behaviour of the BOM (shown in *Figure 3f*) more closely resemble that seen in participants (*Figure 3e*). First, we tested the impact of completely removing the ability of the BOM to adjust to changes in either volatility (*Figure 3b*) or noise (*Figure 3c*) by removing the top nodes of the model (i.e. *kmu* or *vs* respectively). Removing these nodes forces the BOM to estimate the mean volatility or noise across all task blocks rather than estimating local periods where they are higher or lower (see *Animation 1*). As illustrated in *Figure 3g–h*, neither of these lesions recapitulates the pattern of learning rates observed in participants, with the volatility lesioned model attributing increased volatility to noise and thus decreasing its learning rate during periods of higher volatility (main effect of volatility; $F_{(1,696)}=11.9$, $p<0.001$) and the SD-lesioned model treating any form of uncertainty as volatility and thus increasing its learning rate in response to increased noise (main effect of noise; $F_{(1,696)}=227$, $p<0.001$). This suggests that human participants are able to adapt to changes in outcome volatility and noise to some degree, but are less sensitive to these changes than the intact BOM.

We next assessed whether a relative degrading of the model's representation of volatility and noise (*Figure 3d*) altered its behaviour in a manner similar to participants. This was achieved by independently coarsening the model's representation of volatility and noise, with the degree of coarsening selected to make the model's choices as similar as possible to those of a given participant. Details of this coarsening process are provided in the methods section, but in simple terms, at one extreme, the intact model's beliefs about current volatility and noise are represented as probability distributions over many possible values, with the number of values used gradually reduced during coarsening, until the coarsest model treats each form of uncertainty as being either 'high' or 'low.' As can be seen from *Figure 3i*, this relative degrading of the model's representation of uncertainty more closely recapitulated the learning rates observed in participants, with a significant increase in learning rate in response to increased volatility ($F_{(1,696)}=59$, $p<0.001$) and no effect of noise ($F_{(1,696)}=2.3$, $p=0.13$).

In total, the BOM fitted to participant choices had five parameters (i.e. volatility and SD acuity for rewards and losses and a single inverse temperature term). This was compared with two reinforcement learning models: the measurement model described previously, which was fitted to individual blocks and, therefore, had many more parameters (18 in total; learning rates for wins and losses for each block, one inverse temperature term per block), and a simple version of the same model which was fitted across all blocks and had three parameters in total (win and loss LR and one inverse temperature parameter). Model comparison between the BOM and the RW models based on BIC scores favoured the BOM (mean (SD) for BOM; 235 (54), for complex RL measurement model; 281 (57), for simple RL model; 246 (58)).

In the next sections, we characterise how coarsening the BOM changes its behaviour and assess whether it provides an accurate account of participants' noise insensitivity.

## The degraded BOM misattributes noise as volatility

The BOM was degraded by reducing the number of bins it used to represent volatility and/or noise, until its behaviour most closely matched that of participants. This process led to a greater coarsening of the noise than the volatility dimension (*Figure 4a*; F(1,69)=49, p<0.001), with no effect of outcome valence (F(1,69)=0.73, p=0.4), suggesting that the degraded model maintained a generally less precise representation of noise than volatility. In order to investigate the impact of this coarsening on the model's beliefs, we used the degraded BOM's estimates of volatility and noise to categorise task trials as either high or low volatility/noise (i.e. trials in which the model's estimates of these variables were higher/lower than the mean) and compared these to the same trial labels generated by the intact BOM. Consistent with the greater degradation of the noise dimension, coarsening the model caused it to miscategorise more trials which the intact BOM had labelled as having high than low noise (*Figure 4b*; F(1,69)=30.7, p<0.01) with no effect of volatility (F(1,69)=1.9, p=0.17) or outcome valence (F(1,69)=0.004, p=0.95). As illustrated in *Figure 4c*, when the degraded BOM miscategorised high noise trials, it tended to label them as having high, rather than low, volatility. Overall, these results indicate that coarsening the BOM caused it, relative to the intact BOM, to misattribute high noise trials as high volatility trials.

## The degraded BOM rescues optimal behaviour

The process of fitting the degraded BOM to participant behaviour can be understood as searching for a configuration of the model (akin to a grid-based maximum likelihood estimation) in which participant choice conforms to the normative response to volatility and noise coded in the model's structure. In other words, participants' learning rates should increase when the degraded BOM's estimate of volatility is high and, critically, when it estimates that noise is low. We demonstrate this by reanalysing participant behaviour, using the trial labels of the degraded BOM to indicate periods of low/high volatility and noise in place of the task block labels used in the original analysis. In effect, this approach allows us to test whether participants make internally consistent errors during learning, i.e., reducing their learning rates for outcomes that they thought — instead of the actual task structure — were associated with high volatility and/or low noise. As can be seen (*Figure 4f*), participants significantly increased their learning rate when the degraded BOM estimated volatility to be high (F(1,566)=86, p<0.001) and noise to be low (F(1,566)=81, p<0.001). In control analyses, this normative response to uncertainty was not seen when the labels from the intact rather than the degraded BOM were used (*Figure 4e*), or when the BOM's representation of outcome mean was degraded, rather than its estimates of volatility and noise (supplementary materials).

## Assuming human participants use the degraded BOM's estimates of volatility and noise also rescues normative pupillary response

If the degraded BOM is a fair representation of how participants are performing the learning task, then we would expect it to be better able to explain physiological markers of uncertainty estimation than the simple task block structure or the intact BOM. Specifically, participants' pupils should be larger when the degraded BOM thinks that volatility is high and when it thinks noise is low. We first show (*Figure 5a–c*) that participants' pupils do not adapt normatively to the task block structure, with no main effect of block volatility (F(1,1723)=0.002, p=0.9) and an increase of pupil size in response to higher noise (F(1,1723)=13.8 p<0.001). In contrast, analysis using the trial labels derived from the

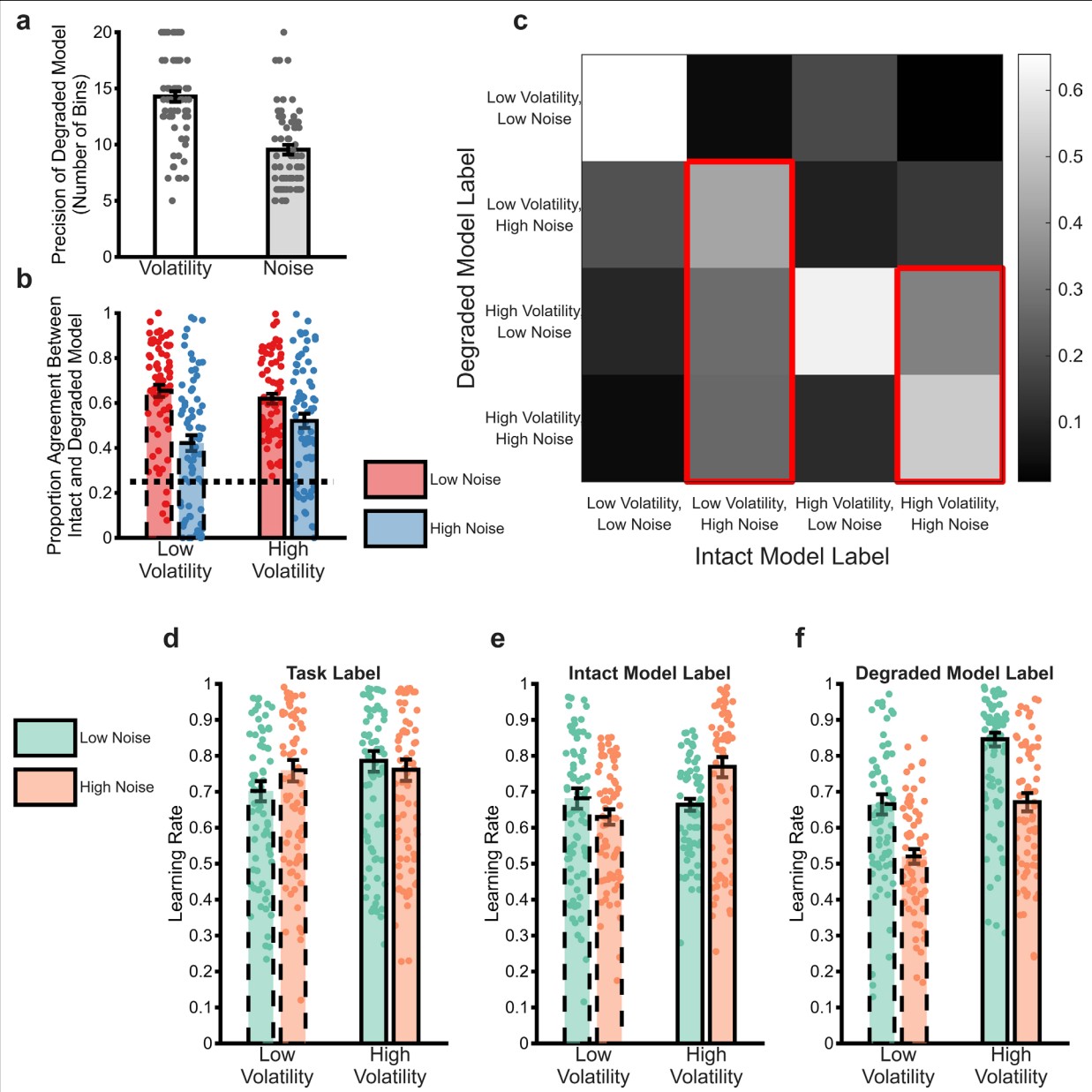

**Figure 4.** Analysis of the behaviour of the degraded Bayesian observer model (BOM). The process of degrading the BOM involved reducing the number of bins used to represent the volatility and noise dimensions independently until the choice of the model matched that of participants. Panel (**a**) illustrates the number of bins selected by this process for the volatility and noise dimensions (averaged across win and loss outcomes). As can be seen, the degraded BOM maintained a less precise representation of noise than volatility. In order to understand the behaviour of the degraded model, the model's estimated $vmu_i$ and $SD_i$ were used to label individual trials as high/low volatility and noise (NB greater than or less than the mean value of the estimates). These trial labels were compared with the same labels from the intact model, which were used as an ideal comparator (panels **b** and **c**). Panel **b** illustrates the proportion of trials in which the labels of the two models agreed, arranged by the ground truth labels of the full model and averaged across win and loss outcomes. The dotted line indicates the agreement expected by chance. The degraded model trial labels differed from those of the full model particularly for high noise trials, with no impact of trial volatility. Panel **c** provides more details on how the degraded model misattributes trials. In this figure, the labels assigned by the full model are arranged along the x-axis. The colour of each square represents the proportion of trials with a specific full model label that received the indicated label of the degraded model (arranged along the y-axis). The diagonal squares illustrate agreement between models as reported in panel (**b**). As highlighted by the red outlines, trials which the full model labelled as having high noise were generally mislabelled by the degraded model as having high volatility. Reanalysis of participant choices using the trial labels provided by the full (panel **e**) and degraded (panel **f**) models indicates that participants adapt their learning rates in a normative fashion when the degraded model trial labels are used (panel **f**), but not when the full model labels are used (panel **e**). Panel (**d**) illustrates the same analysis using the original task block labels for

*Figure 4 continued on next page*

*Figure 4 continued*

comparison. Bars represent the mean (± SEM) of participant learning rates, with raw data points presented as circles behind each bar. See *Figure 4— figure supplement 1* for a comparison of the behaviour of the degraded BOM with an alternative fitted model.

The online version of this article includes the following figure supplement(s) for figure 4:

**Figure supplement 1.** Comparison of the degraded volatility/noise Bayesian observer model (BOM) and a control model ('mu model') in which the representation of the mean of the generative process, rather than the volatility/noise are degraded.

**Figure supplement 2.** Performance of the degraded Bayesian observer model (BOM) on schedules derived from *Nassar et al., 2012*.

degraded model (*Figure 5d–f*) recovered the expected increase in pupil size in response to both raised volatility ($F(1,2067)=105$, $p<0.001$) and reduced noise ($F(1,2067)=42.3$, $p<0.001$) suggesting that the model provides a reasonable measure of participants' estimates of these parameters. Finally, we tested whether the degraded BOM was able to explain more variance in the pupil data than the intact BOM. In order to do this, we first regressed participants' pupil data against the estimated volatility and noise of the intact BOM, as well as a range of other task- related factors (*Figure 5g*; see methods for more details of analysis). Having removed the variance accounted for by these factors, we then regressed the residuals of this first level analysis against the degraded model's estimates of volatility and noise. This second level analysis (*Figure 5h–i*) indicated that the degraded model was able to account for variance associated with outcome noise that was not explained by the full model ($F(1,286)=4.1$, $p=0.04$), but did not explain additional variance associated with outcome volatility ($F(1,286)=0.1$, $p=0.75$). In summary, assuming that participants used the degraded BOM's estimates of outcome volatility and noise rescued the normative pattern of physiological adaptation during the task.

## Discussion

Humans respond in a rational, if approximate, manner to the causal statistics of dynamic environments. We found that participants adapted as expected to changes in outcome volatility, but were relatively insensitive to changes in noise. Using a degraded BOM to characterise participants' behaviour suggested that they responded appropriately to a relatively coarse estimation of the level of noise, that led to its misattribution as volatility. Analysis of pupillometry data using the degraded model again suggested that participants were responding normatively to changes in estimated noise, but that these estimates diverged from the true noise of experienced outcomes. These results illustrate that human learners are able to adapt to the statistical properties of their environment, but during this process, they make internally consistent errors, utilising higher learning rates as a result of misattributing environmental noise as volatility which also leads to suboptimal choice.

Using a task in which volatility and noise varied independently between blocks, we found that human learners adapted as expected (*Behrens et al., 2007*; *Browning et al., 2015*; *Nassar et al., 2012*; *Pulcu and Browning, 2019*) to blockwise changes in the volatility of both win and loss outcomes, increasing the learning rate used when volatility was high vs. low. In contrast, the expected reduction of learning rates in response to increased outcome noise was not apparent, with participants employing a significantly higher learning rate in response to increased noise when volatility was low and a numerically lower learning rate when volatility was high. The absence of a normative response to blockwise changes in noise is at odds with previous work which has described a reduction in learning rates during periods of high noise (*Diederen and Schultz, 2015*; *Nassar et al., 2010*; *Nassar et al., 2012*). However, in this previous work the level of noise was either explicitly presented to participants (as a bar on screen representing the standard deviation of the generative process in Diederen & Schultz) or was made unambiguous by being very different from changes caused by volatility (in Nassar et al, noise was generated using an SD of 5 or 10, while the average change due to volatility was 100). By design, in the current task, high noise and volatility resulted in a similar range of magnitudes (*Figure 1b*) forcing participants to use the temporal sequence of outcomes to discriminate between the different forms of uncertainty. Our behavioural results suggest that, in the absence of unambiguous differences between outcomes caused by volatility and those caused by noise, participants' ability to estimate and/or adapt to changes in noise is reduced. Interestingly, a recent study reported that participants do not adjust their choice or estimated confidence in response to variability in the orientation of arrays of visual gratings (*Herce Castañón et al., 2019*), suggesting

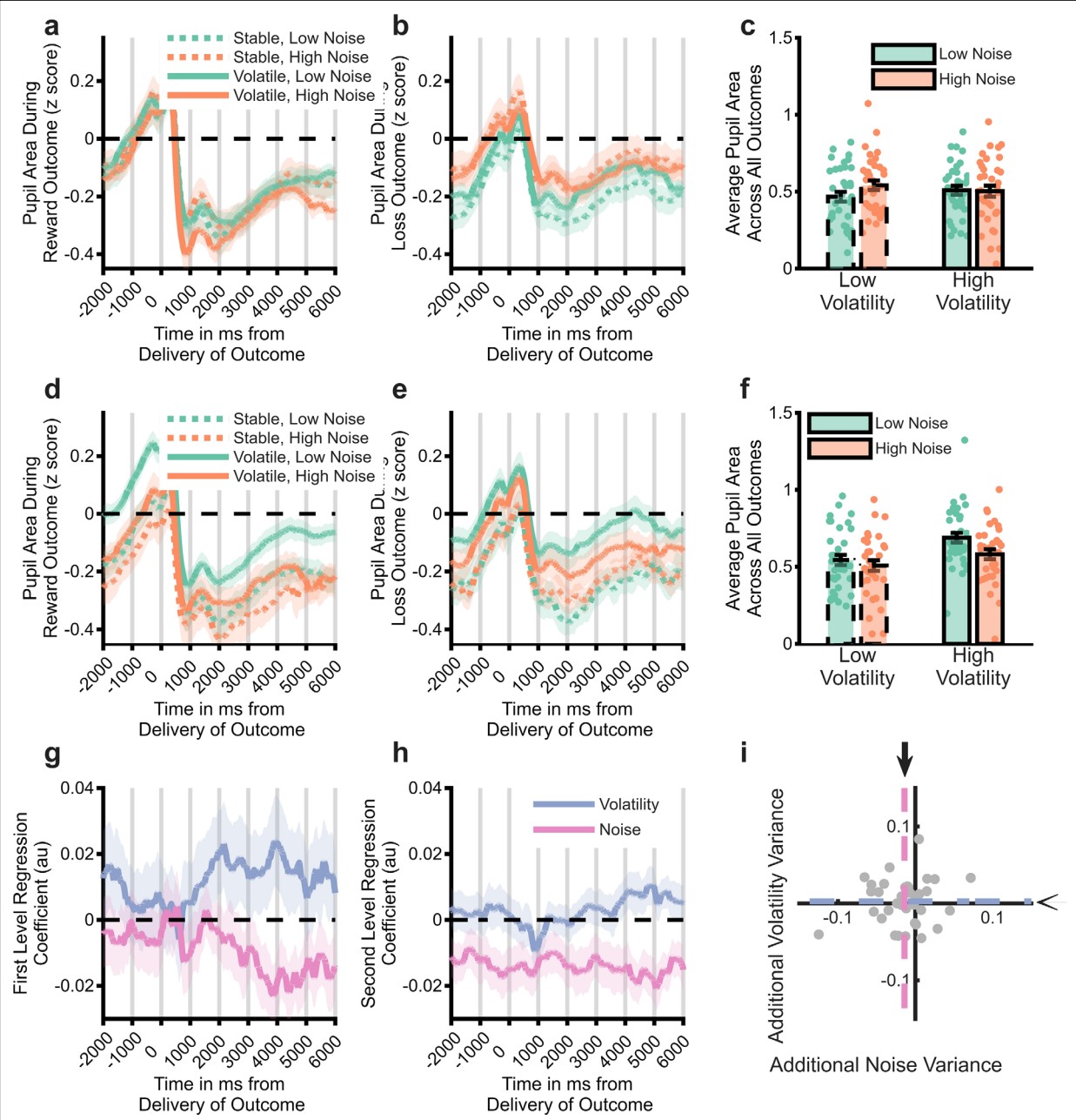

**Figure 5.** Analysis of pupillometry data. Z-scored pupil area from 2 s before to 6 s after win (panel **a**) and loss (panel **b**) outcomes, split by task block. Lines illustrate average size, with shaded area illustrating SEM. Panel **c** Pupil size averaged across whole outcome period and both win and loss outcomes. Pupil size did not systematically vary by task block. Panels (**d-f**) as above but using the trial labels derived from the degraded model. Pupil size was significantly larger for trials labelled as having high vs. low volatility and low vs. high noise. Panel (**g**) displays the mean (SEM) effect of volatility and noise as estimated by the full BOM derived from a regression analysis of pupil data. The residuals from this analysis were then regressed against the estimated volatility and noise from the degraded model. A time course of the regression weights from this analysis is shown in panel **h**, with the mean coefficients across the whole period shown in panel **i**. The degraded model's estimated noise accounted for a significant amount of variance not captured by the full model (pink line in **h** is below 0, the mean effect across the period is represented by dashed lines and arrows in panel **i**). See *Figure 5—figure supplement 1* for comparison of the degraded BOM with an alternative fitted model.

The online version of this article includes the following figure supplement(s) for figure 5:

**Figure supplement 1.** Comparison of the degraded volatility/noise Bayesian observer model (BOM) and the mu BOM on analysis of the pupillometry data.

that an insensitivity to outcome noise may be a general feature of human decision making, rather than a specific component of learning.

Noise fundamentally limits the reliability of information (*MacKay, 2003*) and ignoring it has a clear detrimental impact on inference (*Figure 3h*), causing agents to be unnecessarily influenced by chance events (*Pulcu and Browning, 2019*). It would, therefore, be surprising if human learners were completely insensitive to this process, particularly given evidence that they can respond normatively when the level of noise is unambiguous (*Diederen and Schultz, 2015*; *Nassar et al., 2010*; *Nassar et al., 2012*). We developed an ideal (BOM; *Behrens et al., 2007*; *Nassar et al., 2010*; *Piray and Daw, 2021*; *Pulcu et al., 2022*) to investigate the degree to which participants were adapting to noise. The intact BOM displayed the expected behavioural response to changes in both volatility and noise (*Figure 3f*) and, as a result, did not accurately capture the behaviour of participants (*Figure 3e*). Completely removing the BOM's ability to adapt to noise (or volatility) did not recapitulate participant choice behaviour (*Figure 3g–h*), whereas coarsening its representation of volatility and noise, produced a much closer match (*Figure 3i*). This suggests that participants were relatively, rather than completely insensitive to noise and that they tended to misattribute high noise as volatility (*Figure 4*). However, an important caveat to this interpretation is that the degree of coarsening was selected using participants' choices. The better behavioural match of the coarsened BOM to participant learning rates may, therefore, be simply because this model was fitted to the same choices used to calculate the learning rates, whereas the intact and fully lesioned models were not. We, therefore, sought to validate the coarsened BOM by assessing its ability to account for participants' pupillary data, and by comparing it with an alternative fitted BOM which coarsened the representation of the generative mean, rather than the estimated uncertainty (see Supplementary materials). Participants' pupil size did not vary systematically between different block types, whereas they were significantly larger when the degraded BOM estimated volatility to be high and noise to be low (*Figure 5a–f*). Similarly, the estimated noise of the degraded BOM accounted for additional variance in pupil size, over and above the intact BOM (*Figure 5g–i*). In contrast, the alternative mean-degraded BOM did not recapitulate participants' learning rates (*Figure 4—figure supplement 1*) and was not able to account for changes in participant pupil size (*Figure 5—figure supplement 1*). The finding that participants' pupil size covaries in the expected direction with the degraded BOM's estimated levels of both volatility and noise provides some reassurance that the model is capturing the dynamics of participants' uncertainty estimates. More generally, the presence of both volatility and noise signals in this data, indicate that, as suggested previously (*Nassar et al., 2012*; *O'Reilly et al., 2013*), the pupillometry signal reflects general belief updating rather than specifically volatility.

An outstanding question is why participants might be particularly insensitive to changes in outcome noise. It is tempting to try to answer this question by reference to the processes by which the BOM was coarsened (i.e. the insensitivity was caused by a reduction in the precision by which noise was represented in a multi-dimensional probability distribution). However, the BOM described here was developed as an algorithmic description of how the learning task may be solved. As far as we are aware, there is little evidence that it accurately describes the cognitive or neural implementation of uncertainty estimation. Alternative algorithmic approaches to the general problem of uncertainty estimation have been described (*Kalman, 1960*; *Nassar et al., 2010*; *Piray and Daw, 2021*; *Pulcu and Browning, 2019*), including simpler approaches that avoid computationally expensive representations of multi-dimensional distributions (*Kalman, 1960*; *Nassar et al., 2010*) and which, therefore, may be more likely implementational candidates. In other words, the current results indicate that human learners are relatively insensitive to changes in outcome noise, but do not specify the lower level mechanisms that determine this effect.

Previous work examining the neural representations of uncertainty has tended to report correlations between brain activity and some task-based estimate of one form of uncertainty at a time (*Behrens et al., 2007*; *Walker et al., 2020*; *Walker et al., 2023*). We are not aware of work that has, for example, systematically varied volatility and noise and reported distinct correlations for each. An interesting possibility as to how different forms of uncertainty may be encoded is suggested by parallels with the neuronal decoding literature. One question addressed by this literature is how the brain decodes changes in the world from the distributed, noisy neural responses to those changes, with a particular focus on the influence of different forms of between-neuron correlation (*Averbeck et al., 2006*; *Kohn et al., 2016*). Specifically, signal-correlation, the degree to which different

neurons represent similar external quantities (required to track volatility) is distinguished from, and often limited by, noise-correlation, the degree to which the activity of different neurons covaries independently of these external quantities. One possibility relevant to the current study, which resembles the underlying logic of the BOM, is that a population of neurons represents the estimated mean of the generative process that produces task outcomes. In this case, volatility would be tracked as the signal-correlation across this population, whereas noise would be analogous to the noise-correlation and, crucially, misestimation of noise as volatility might arise as misestimation of these two forms of correlation. While the current study clearly cannot adjudicate on the neural representation of these processes, our finding of distinct behavioural and physiological responses to the two forms of uncertainty, does suggest that separable neural representations of uncertainty are maintained.

A related question is whether other, non-Bayesian model formulations may be able to account for participants' learning adaptation in response to volatility and noise. Of note, the reinforcement learning model used to measure learning rates in separate blocks does not achieve this goal—as this model is fitted separately to each block rather than adapting between blocks (NB the simple reinforcement learning model that is fitted across all blocks does not capture participant behaviour, see supplementary information). One candidate class of model that has potential here is latent-state models (*Cochran and Cisler, 2019*), in which the variance and unexpected changes in the process being learned (which have a degree of similarity with noise and volatility, respectively) is estimated and used to alter the model's rates of updating as well as the estimated number of states being considered. Using the model described by Cochran and Cisler, we were unable to replicate the learning rate adaptation demonstrated by participants in the current study (see Supplementary information), although it remains possible that other latent state formulations may be more successful.

In conclusion, human learners adapt rationally to estimates of the volatility and noise of experienced outcomes. However, these estimates are approximate leading to a relative insensitivity to outcome noise.

## Methods
## Experimental model and subject details
### Participants
70 English-speaking participants aged between 18 and 65 were recruited from the general public using print and online advertisements. A previous study (*Pulcu and Browning, 2017*) on behavioural response to changes in volatility reported an effect size of $d$=0.7. As the effect size of a noise manipulation was not clear, we recruited a sample size sufficient to detect an effect size of half this value ($d$=0.35) with 80% power. Participants were excluded from the study if they had any psychological or neurological disorders or were currently on psychotropic medication. No exclusion criteria related to task performance were used.

## Method details
### General procedure
Participants attended a single study visit during which they completed the learning task. The study was approved by the University of Oxford Central Research Ethics Committee (R49753/RE001). All participants provided written informed consent to take part in the study, in accordance with the Declaration of Helsinki.

### Behavioural paradigm
The reinforcement learning (RL) task consisted of six blocks, each comprising 60 trials. In each trial, participants were presented with two abstract shapes taken from the Agathodaimon font (i.e. shape A and shape B). Two different shapes were used in each block, with rest sessions between blocks. The shapes were presented randomly on either side of the screen. Participants were explicitly instructed that this randomised location did not influence the outcome magnitudes. Participants attempted to accumulate as much money as possible by learning the likely magnitude of the wins and losses associated with each shape and using this information to guide their choice. On each trial, participants chose one of two shapes, with their choice highlighted by a black frame (see *Figure 1a*). Following the choice, the win and loss amounts associated with the chosen shape were presented, in randomised

order, for a jittered period (2–6 s, mean: 4 s) inside two empty bars, above and below the fixation cross. The win amount was shown as a green area in the upper bar, and the loss amount was represented as a red area in the lower bar. The total length of each bar represented £1 (i.e. of wins or losses) and thus the amount associated with the chosen shape was the proportion of the bar filled by the green/red areas (e.g. three quarters of the upper bar being green, would mean that the chosen option was associated with a win of 75 p). Participants were informed that the unshaded area of each bar was the amount associated with the unchosen option. Thus, on each trial, participants knew how much they had won/lost and how much they would have won/lost if they had chosen the other option. This feature simplified the task; rather than having to separately estimate the wins and losses associated with each shape, participants only had to estimate these values for one shape (with the other shape being £1 minus this value). For each trial, participants received the difference between the win and loss amounts associated with their choice. A running total amount of money was displayed in the centre of the screen, under the bars, and was updated at the beginning of the subsequent trial with the recent winnings. Participants were informed that the task would be split into six blocks, that they had to learn which was the best option to choose, and that this option may change over time. They were not informed about the different forms of uncertainty we were investigating or of the underlying structure of the task (that uncertainty varied between blocks).

The wins and the losses associated with each shape followed independent outcome schedules (*Figure 1b*), generated from a Gaussian distribution. In each block, the win and loss outcomes had either high or low volatility and high or low noise. When volatility was low, the mean of the Gaussian distribution remained constant, when volatility was high, the mean changed between 25–40, and 60–75 every 9–15 trials. When noise was low, the standard deviation of the Gaussian was set to 5, whereas when noise was high, the standard deviation was 35. As can be seen from *Figure 1b*, these schedules resulted in similar ranges of outcome magnitudes for periods of high noise and high volatility. The first block for every participant had high volatility and low noise for both win and loss outcomes and was used to familiarise participants with the task. Choices from this block were not used in the analyses presented (although including them does not alter the reported pattern of results). The schedules in the remaining five blocks were presented in a randomised order with the constraint that, across both win and loss outcomes, each of the four combinations of volatility and noise level (*Figure 1B*) was presented either 2 or 3 times. Thus, while each participant completed at least two blocks with each of the four combinations of high/low volatility/noise, the specific pairings of win and loss volatility/noise levels, differed across participants. This approach was used in preference to a fully factorial design in order to keep the total task duration to a manageable level. At the end of the experiment, participants were paid one-fifth of their total winnings, plus a £15 baseline rate for turning up to take part.

Pupillometry data was collected for 36 of the 70 participants. During the collection of pupillary data, the task was presented on a VGA monitor connected to a laptop computer running Presentation software version 18.3 (Neurobehavioural Systems). An identical behavioural version of the task, presented using Psychtoolbox 3.0 on MATLAB (MathWorks Inc), was used to collect behavioural data from the remaining 34 participants. In the pupillometry version, participants' heads were stabilised using a head-and-chin rest placed 70 cm from the screen on which the eye tracking system was mounted (Eyelink 1000 Plus; SR Research). The eye tracking device was configured to record the coordinates of both of the eyes and the pupil area at a rate of 500 Hz. The task stimuli were drawn on either side of a fixation cross which marked the middle of the screen and were offset by 7° visual angle. The testing session lasted approximately 70 min per participant.

## Analysis of choice data
### Non-model-based measure of the influence of outcomes
The manipulation of uncertainty in the reinforcement learning task is expected to alter the degree to which participants' choices are influenced by the outcomes they experience. A simple, if somewhat crude, measure of this influence can be calculated as the proportion of trials in a block in which participants select the choice prompted by the win or loss outcomes on the previous trial. Generally, win outcomes of >50 p and loss outcomes of <50 p associated with a shape will prompt selection of the same shape on the next trial, whereas other outcomes will prompt selection of the alternative shape. The overall effect of win outcomes on choice can, therefore, be estimated as:

$$P_{(choice==A \mid previous\ win\ outcome\ for\ A>50p)} - P_{(choice==A \mid previous\ win\ outcome\ for\ A<50p)}$$

That is, the probability of choosing shape A, given that, on the previous trial, a win of >50 p was associated with shape A – the probability of choosing Shape A, given that, on the previous trial a win of <50 p was associated with Shape A. Similarly, the effect of loss outcomes is estimated as:

$$P_{(choice==A \mid previous\ loss\ outcome\ for\ A<50p)} - P_{(choice==A \mid previous\ loss\ outcome\ for\ A>50p)}$$

However, choice is also influenced by the magnitude of the outcome; a win of 90 p will have a greater effect on subsequent choice than a win of 55 p. Blocks with high levels of either volatility or noise have more extreme magnitudes than blocks with low levels of both (*Figure 1b*) which will bias any comparison of this metric between blocks. In order to limit the effect of this bias, we estimated the simple choice metric only for trials in which the previous outcome lay in the range of magnitudes common to all four blocks, 35–65.

## Reinforcement learning model

While the choice metric described above provides a relatively transparent measure of the influence of task outcomes on choice, it does not account for differences in outcome magnitude making it liable to bias. We, therefore fitted a simple reinforcement learning model to measure block-wise learning rates, which provide a more principled estimate of the degree to which choices are influenced by outcomes. The model combines a learning phase in which the magnitude of wins and losses associated with a shape are estimated (note that it is not necessary to learn the magnitudes associated with the other shape, as these are simply 1- those described below)

$$Qwin\_a_{(t+1)} = Qwin\_a_{(t)} + \alpha_{win}\left(win_{(t)} - Q_{win\_a(t)}\right)$$

$$Qloss\_a_{(t+1)} = Qloss\_a_{(t)} + \alpha_{loss}\left(loss_{(t)} - Q_{loss\_a(t)}\right)$$

In these equations, $Qwin\_a_{(t)}$ and $Qloss\_a_{(t)}$ are the estimated win and loss magnitudes associated with Shape A on trial $t$, $win_{(t)}$ and $loss_{(t)}$ are the observed win and loss outcome magnitudes and $\alpha_{win}$ and $\alpha_{loss}$ are the win and loss learning rates. These values are then combined in a decision phase such that:

$$Pchoice\_a_{(t)} = \frac{1}{1 + e^{-\beta\left(Q_{win\_a(t)} - Q_{loss\_a(t)}\right)}}$$

where $Pchoice\_a_{(t)}$ is the probability that Shape A will be chosen on trial $t$ and $\beta$ is a single inverse decision temperature. This model was initiated with $Qwin\_a_{(0)} = Qloss\_a_{(0)} = 0.5$ and the three free parameters ($win_{(t)}$, $loss_{(t)}$, and $\beta$) were estimated for each block and each participant by calculating the joint posterior probability given participant choice, marginalising each parameter, and deriving the parameters' expected values (*Behrens et al., 2007*; *Browning et al., 2015*). See supplementary materials for model selection data.

Some analyses reported in the paper (i.e. where trials are labelled as high/low volatility and high/low noise by the Bayesian Observer Model rather than by task block) cannot be modelled using this block-wise approach (as different types of trials are interleaved throughout the task, rather than blocked). In these analyses, a similar single model was fit across all trials in the task. This model had eight different learning rates (separate win and loss learning rates, for each combination of high/low volatility and high/low noise labelled trials) and a single inverse temperature parameter. Although this model is somewhat less flexible than the blockwise modelling approach (i.e. it has 8, rather than 10 learning rates, and 1 rather than 5 inverse temperatures), it produces the same pattern of results when applied to participant choices split by task block (all estimated learning rates correlate at $r>0.8$, *Figure 2c–d* show results from blockwise fitting, *Figure 3e* from the simpler model). This simpler model was fit using stan, with 5000 burn in and 5000 estimation trials, with posterior convergence visually checked and rhat values of less than 1.1 accepted.

Note that neither of these models describe how participants adjust to different levels of volatility and noise, they simply estimate the learning rates used in each block/type of trial, which are expected

to vary in response to differences in levels of uncertainty (in contrast, the Bayesian Observer Model described below does estimate uncertainty and adjust to levels of uncertainty).

## Bayesian observer model

A recursive, grid-based BOM was developed, similar to that described by Behrens and colleagues (*Behrens et al., 2007*; *Pulcu et al., 2022*). The BOM is based on a generative process (see *Figure 3*), and described fully in *Pulcu et al., 2022*. Below, we summarise the key aspects of the model.

The BOM assumes that the observed outcomes at a given time point $t$, $y_t$, are generated from a Gaussian distribution with an unknown mean, $\mu_t$, and standard deviation, $e^{SD_t}$, with the later producing noise in the observed outcomes (*Figure 1b–c*).

$$y_t \sim N\left(\mu_t, e^{SD_t}\right)$$

As illustrated in *Figure 1b–c*, the mean of this distribution may change between time points, leading to volatility in the task environment, with this change described by a second level Gaussian distribution, centered on the current mean and with a standard deviation of $e^{vmu_t}$. The mean of the generative Gaussian distribution in the following trial is drawn from:

$$P\left(\mu_{t+1}\right) \sim N\left(\mu_t, e^{vmu_t}\right)$$

Both the noise ($SD_t$) and volatility ($vmu_t$) parameters can also change between time points with their change governed by Gaussian distributions centered on their current value with standard deviations of $e^{vSD}$ and $e^{kmu}$, respectively. These higher-level parameters allow the model to account for periods in which noise and volatility are high and other periods in which they are low (for example, as caused by the uncertainty changes between task blocks).

$$P\left(vmu_{t+1}\right) \sim N\left(vmu_t,\ e^{kmu}\right)$$

$$P\left(SD_{t+1}\right) \sim N\left(SD_t, e^{vSD}\right)$$

The BOM estimates the joint posterior probability of the five causal parameters, given the choice outcome it has observed. The joint probability distribution at time point $t$ is defined as:

$$P\left(joint_t\right) = P\left(mu, vmu, kmu, SD, vSD | y_{t-1}, y_{t-2},\ \ldots, y_1\right)$$

This joint probability distribution can be thought of as the BOM's belief about the values of each parameter in the generative model. A Markovian assumption (i.e. that nodes of the model are sufficient to describe the generative process) simplifies this process and illustrates the recursive update performed by the BOM:

$$P\left(joint_t\right) = P\left(mu_t, vmu_t, kmu_t, SD_t, vSD_t \mid joint_{t-1},\ y_{t-1}\right)$$

We initialized the joint posterior, before observation of any task outcomes as a uniform distribution. The BOM performs the update, first using Bayes' rule to incorporate the effect of the most recently observed outcome, and then accounts for the drifting parameters by using the conditional probability of the new value of the drifting parameter, given the initial value and drift rate (See; *Pulcu et al., 2022* for a detailed account of this updating process):

$$\left(joint_t \mid joint_{t-1}, y_{t-1}\right) = \iiint p\left(joint_{t-1} \mid y_{t-1}\right) p\left(SD_t \mid SD_{t-1}, vSD\right) p\left(vmu_t \mid vmu_{t-1}, kmu\right) \ldots$$

$$p\left(mu_t \mid mu_{t-1}, vmu_t\right),\ dSD_{t-1},\ dvmu_{t-1},\ dmu_{t-1}$$

The value of each node is derived at every time point by marginalizing over all but the relevant dimension of the joint probability distribution and calculating the expected value of that dimension.

During the task, the shapes presented to participants change between each task block, which means that, at the start of each block, participants have to relearn the mean associated with each shape. This was dealt with in the BOM by flattening the mu dimension of the joint probability distribution at the start of each trial (i.e. replacing the values of the mean dimension, with the average of

the joint distribution across this dimension). The effect of this is to reset the model's belief about the actual magnitude associated with the two new shapes, while maintaining its belief about the overall volatility and noise of the outcomes.

The BOM was provided with the win and loss outcomes (as values between 0 and 1) for each trial, across all trials in the task (excluding the first practice block, although including this did not alter the pattern of results). It treated the two outcomes as independent (i.e. the win outcome did not influence estimates for the loss outcome and vice versa) and transformed the outcomes to the infinite real line using the logistic transform before estimating the posterior probability (*Pulcu et al., 2022*).

### Lesioning the Bayesian observer model

A number of different lesions were applied to the BOM. First, its ability to estimate changes in either volatility or noise was removed. This was achieved simply by removing the kmu or vSD nodes from the BOM (reducing the dimensionality of the joint distribution by one in each case). The effect of this is to force the BOM to estimate the mean volatility and noise (respectively) across the whole task, rather than to modify its estimates of these parameters between trials.

The second approach induced a graded, rather than absolute, lesion. This was achieved by reducing the precision with which the BOM represented the volatility-related nodes (vmu and kmu) and/or the noise- related nodes (SD and vSD). More specifically, the BOM's estimates of the values of each of the five nodes are encoded on a five-dimensional grid, with each dimension on the grid representing the possible range of values of a particular node, from low to high, using a fixed number of points. The probability ascribed by the model to a specific point on this dimension is the relative probability that the value of the node lies within the bin of values that is closer to the point, than to adjacent points. For example, say the value of volatility (vmu) ranged from 0 to 10 and was represented by 10 bins. In this case, volatility would be represented by a probability mass function over the 10 bins (<0.5, 0.5–1.5, 1.5–2.5, …,>9.5). Lesioning occurred by independently varying the number of bins used in the volatility-related and/or noise-related dimensions, from a maximum of 20, to a minimum of 2 (i.e. with only two bins, volatility/noise would be represented as simply 'high' or 'low'). The degree of lesioning selected for each individual participant was determined as the number of bins for the volatility and noise dimensions that, after passing the model estimates through a softmax action selector with a single inverse temperature parameter (i.e. as described for the RL model), maximized the likelihood that the model would make the same choices as the participant, across all task blocks. This process of lesioning, therefore, progressively coarsens the BOM's representation of the two types of uncertainty and selects the degree of coarsening that results in choices as similar as possible to participants (see supplementary materials for an alternative model that coarsens the representation of the mean values).

### Alternative measurement model

A reinforcement learning model with separate learning rates for win and loss outcomes and a single beta term was used to estimate the learning rates employed by participants and the BOMs in the paper. An alternative, slightly more complex version of this model, uses separate learning rates and separate beta terms for the two outcomes. This model estimates Q values in a similar manner, but uses the following approach during action selection:

$$Pchoice\_a_{(t)} = \frac{1}{1 + e^{-\left(\beta_{win} * \left(Q_{win_{a(t)}} - 0.5\right) - \beta_{loss} * \left(Q_{loss_{a(t)}} - 0.5\right)\right)}}$$

As can be seen, two separate beta terms are used to separately weight the win and loss Q values when selecting an action.

This more complex model provided a poorer fit to participant choice data (mean AIC/BIC: 40.9/41.9) compared to the simpler model with a single beta term (mean AIC/BIC: 39.6/40.4). Furthermore, as illustrated in *Figure 2—figure supplement 1*, the learning rates recovered from the more complex model show the same pattern of effects when analysing participant behaviour as those demonstrated by the simpler model (main effect of volatility: $F_{(1,696)}=47.4$, $p<0.001$, main effect of noise: $F_{(1,696)}=0.37$, $p=0.54$, interaction between volatility and noise $F_{(1,693)}=5.24$, $p=0.02$). In summary,

the simpler model provides a better fit to choice data and provides similar estimates of participant learning rates than the more complex model, therefore, the simpler model was used in the paper.

## Generate-recover performance of the measurement model

The ability of the measurement model to recover the three parameters it encodes (win learning rate, loss learning rate and inverse temperature) was assessed by generating synthetic choices across a range of learning rates (0.01–0.99) and inverse temperatures (1-36; NB the mean recovered beta value was 18) from a single task block and then comparing the recovered values to those used to generate the choices. These results are summarised in *Figure 2—figure supplement 2* . As can be seen, all model parameters are recovered well unless the inverse temperature parameter was very low (i.e. when the choice is made relatively randomly).

## Effect of uncertainty manipulation on inverse temperature

The effects of uncertainty on estimated learning rates are reported in the Results section. Here, we describe the effects of the uncertainty manipulation on estimated inverse temperature (the beta parameter from the estimation model). This analysis was run as for the learning rate analysis: log-transformed beta parameters were entered into a linear mixed model with fixed factors of win volatility, win noise, loss volatility and loss noise, and random intercepts for subjects. The estimated choice inverse temperature was lower when the noise of either outcome was higher (win: $F(1,345)=25.3$, $p<0.001$; loss: $F(1,345)=55.7$, $p<0.001$) and was not affected by outcome volatility ($p>0.05$). Including inverse temperature as a covariate in the analysis of participant learning rates did not influence the reported pattern of results.

## Analysis of a control, fitted Bayesian observer model

The degraded BOM was fit to participant choice, with the number of bins used to represent volatility and noise selected to maximise the degree to which model choice matched participant choice. We report that (a) the degraded model adjusts its learning rate in response to changes in uncertainty in a similar manner to participants, (b) that if we label trials as being high/low volatility/noise based on the internal estimates of the degraded model we are able to recover both a normative pattern of behaviour and pupil response from participant data. We use these results to argue that the degraded model provides useful information about how participants estimate and adjust to changes in uncertainty.

In the following section we test whether the ability of the degraded model to produce these results depends on how it represents uncertainty (i.e. changes to the volatility/noise nodes) or whether a similar effect is produced by degrading its estimation of the other node which influences its behaviour, the mean of the generative process (i.e. *mu*). The degrading of the *mu* node was achieved as described in the main text for the volatility/noise, the number of bins used to represent the node were varied, to maximise the likelihood of the model producing the same choice as participants. *Figure 4—figure supplement 1* summarises the analysis of the behavioural data, comparing it to the behaviour of participants (panel a) and the volatility/noise model described in the main paper (panels b & d). As can be seen, whereas the volatility/noise degraded model (panel b) replicates participants' response to changes in uncertainty (panel a), the fitted mu model does not. After fitting to participant behaviour, the mu model uses generally lower learning rates than participants, and specifically shows a higher learning rate when volatility increases ($F(1,696)=84.5$, $p<0.001$), but also a lower learning rate when noise is increased ($F(1,696)=3.9$, $p=0.049$). As described in the main paper, participants do not show this expected reduction in learning rates when noise is raised.

Similarly, unlike the degraded volatility/noise model, using the mu model's estimates of volatility and noise to label trials and then reanalysing participant data did not produce normative behaviour, with an interaction found between volatility and noise ($F(1,537)=7.2$, $p=0.008$) arising from a significantly lower learning rate with higher noise when volatility was high ($F(1,277)=11.4$, $p<0.001$) and a non-significant increase in learning rates with higher noise when volatility was low ($F(1,277)=0.007$, $p=0.93$; Figure panel e).

Next, we assessed the performance of the mu model on the analysis of the pupillometry data, and specifically, whether it explained variance in this data over and above the unfitted, full model. As illustrated in *Figure 5—figure supplement 1*, whereas the degraded volatility/noise model explained

extra variance associated with its estimate of noise (see main *Figure 5*), the mu model did not explain additional variance in this data at all (additional effect explained by volatility $F_{(1,286)}=0.01$, p=0.9; additional effect explained by noise $F_{(1,286)}=0.38$, p=0.54).

Overall, the ability of the degraded model in the main paper to account for participant choice behaviour and changes in pupil size are not replicated when the fitting process influences a non-uncertainty related node (*mu*).

## Does the degraded BOM replicate the effect of noise reported in *Nassar et al., 2012*?

In the main text, we suggest that *Nassar et al., 2012* observed an increase in learning rate during low relative to high noise trials because in their schedule, noise produced a significantly smaller effect on outcomes than volatility (*Figure 4—figure supplement 2*). If this is the case, then we might expect the degraded BOM used in the current study to also show the appropriate learning rate adaptation to changes in noise, if presented with the schedules used in Nassar. *Figure 4—figure supplement 2* illustrates the results of this analysis, which indicated that the degraded BOM did indeed show an increase in learning rate in low relative to high SD blocks ($t(69)=3.6$, p=0.0006). Consistent with the effects reported in the main paper we found that the number of bins used by the degraded model to represent noise was positively associated with the degree to which the model adapted it learning rate in the expected direction (controlling for the bins used to represent volatility), $rparital_{parital} = 4.1$, p<0.001, while the number of bins used to represent volatility was not significant, $rparital_{parital} = -0.04$, p=0.16.

## Performance of an alternative, latent state model

An alternative approach to the estimation of volatility and noise is provided by latent state models (*Cochran and Cisler, 2019*). We assessed the degree to which the online general latent-state model described by *Cochran and Cisler, 2019* was able to account for participant behaviour. Specifically, we assessed (a) the degree to which it captured participant choice, parameterised as blockwise learning rate in the task (see *Figure 3*, main text) and (b) whether internal model estimates of volatility and noise from the latent-state model were able to rescue normative behaviour, as described for the Bayesian Observer Model in the main paper (main paper *Figure 4f*). The latent state model assumes that observations are generated from one of a series of latent states. The model estimates expected uncertainty, qualitatively similar to noise, at each time point as the expected value of the square of the prediction error. It also estimates unexpected uncertainty, which is similar to volatility, as a function of the likelihood ratio between a one-state model and its current prediction. When this likelihood ratio exceeds a threshold (i.e. the unexpected uncertainty is judged to be high), the model creates an additional latent state. The model is described in detail in *Cochran and Cisler, 2019*. 11 model parameters were allowed to vary when fitting the model to participant choice (*Table 2*):

The model's value estimates were reset at the start of each task block, with the number of latent states at the start of each block being set to 1. The number of active latent states was used as an estimate of volatility, the log of the model's expected value for the square of the prediction error was used as an estimate of noise. Model-derived trial labels (i.e. high/low volatility and noise) were calculated as for the analyses in *Figure 4* of the main paper-- those trials with values above/below the mean value for that participant.

**Table 2.** Summary of free parameters in latent state model (see *Cochran and Cisler, 2019* for detailed description).

| Parameter | Description | Separate for win and loss outcomes | Number of parameters |
|---|---|---|---|
| Alpha0 | Learning rate for the association | yes | 2 |
| Alpha1 | Learning rate for variance | yes | 2 |
| Alpha2 | Learning rate for covariance | yes | 2 |
| Gamma | Transition probability between states | yes | 2 |
| Eta | Threshold for creating new state | yes | 2 |
| Beta | Inverse choice temperature | No | 1 |

Figure 3—figure supplement 1 illustrates the estimated learning rates of the latent-state model in the task blocks. As can be seen, it replicates the increased learning rate in high volatility blocks demonstrated by participants ($F_{(1,696)}=9.98$, $p=0.001$), however, unlike participants, it significantly increases its learning rate in response to noise ($F_{(1,696)}=100$, $p<0.001$). Using the labels for high/low volatility and noise derived from the latent-state model (Figure 3—figure supplement 1) does not rescue normative behaviour in participants. Participants employ a higher learning rate in trials the model considers to have low relative to high volatility ($F_{(1,556)}=7$, $p=0.008$), with no effect of noise ($F_{(1,556)}=3.1$, $p=0.08$). These results suggest that, while the current latent-state model is able to capture some aspects of participant behaviour, the internal model estimates of unexpected and expected uncertainty (as markers of volatility and noise, respectively) do not explain the response to uncertainty accounted for by the Bayesian Observer Model. It should be noted that, while the latent state model was constructed to respond to levels of uncertainty (Cochran and Cisler, 2019) the formulation of these, and particularly of unexpected uncertainty is somewhat different to that used in the BOM (e.g. the estimate of unexpected uncertainty is a measure of the degree to which existing latent states are unable to account for experienced outcomes, and it can only increase across a block). It, therefore, remains possible that alternative latent state formulations would be more sensitive to the behaviour examined here.

## Performance of a simple RL model

A final possibility considered is that some of the behaviour of the fitted BOM might be captured by a radically simpler model that does not represent levels of uncertainty. To assess this, we fitted the simple reinforcement learning model to all of a participant's choices across all task blocks (i.e. as compared to the measurement RL model used in the main paper which was fit to individual blocks). We then estimated the effective learning rate per block using choices derived from this model and the same analysis pipeline as the main paper. As can be seen in Figure 2—figure supplement 3, this very simple model does not replicate the learning adaptation apparent in participant behaviour.

## Pupilometry data preprocessing

Pupilometry data were collected using the Eyelink II system (SRresearch) from both eyes, sampled at 500 Hz. Preprocessing involved the following steps: Eye blinks were identified using the built-in filter of the Eyelink system and were removed from the data. A linear interpolation was implemented for all missing data points (including blinks). The resulting trace was subjected to a low pass Butterworth filter (cut-off of 3.75 Hz), z-transformed across the session (Browning et al., 2015; Nassar et al., 2012), and then averaged across the two eyes. The pupil response to the win and the loss outcomes were extracted separately from each trial, using a time window based on the presentation of the outcomes. This included a 2 s pre-outcome period, and a 6 s period following outcome presentation. Individual trials were excluded from the pupilometry analysis if more than 50% of the data from the outcome period had been interpolated (mean = 6.7% of trials) (Browning et al., 2015). The first five trials from each block were not used in the analysis as initial pupil adaptation can occur in response to luminance changes in this period (Browning et al., 2015; Nassar et al., 2012). The preprocessing resulted in two sets of timeseries per participant, one set containing pupil size data for each included trial when the win outcomes were displayed and the other when the loss outcomes were displayed. These pupil area data were binned into 1 s bins across the outcome period for analysis (NB Figure 5a–f). This analysis was supplemented by an individual regression approach (Figure 5g–i) in which individual participants' pupil area timeseries was first regressed against estimated trialwise volatility and noise from the intact BOM (Figure 5g), as well as a number of control variables (constant term, amount won/lost on trial (i.e. magnitude of outcome), valence of outcome (win or loss), order in which outcomes were presented (win first/loss first), trial number (1:360), whether shape chosen switched on next trial or not (1:0)). The residuals from this regression were then regressed against estimated trial-wise volatility and noise from the degraded BOM (Figure 5h and i). These regression analyses resulted in timeseries of beta-weights that were analysed in the same manner as raw pupil size data.

## Quantification and statistical analysis

Behavioural data were analysed using linear mixed effect models (fitlme function of Matlab (2022a)) with participant ID included as a random factor and volatility, noise, and valence added as fixed factors.

Two-way interactions between fixed effects were also tested (main effects are reported from models without interaction terms). Addition of random slopes for any of the fixed factors decreased LME model fit statistics and so were not included (*Matuschek et al., 2017*). Analysis of timeseries pupillometry data included the additional fixed effect factor of time across the outcome period. Learning rates were transformed to the infinite real line using a logistic transform before analyses (untransformed data are displayed in figures for ease of interpretation). The normality of the distribution of the residuals of the LME analyses was checked both visually and with a one-sample Kolmogorov-Smirnov test. Changes in the classification of trials between the full and degraded BOM (*Figure 4b*) were analysed using a repeated measures ANOVA with within-subject factors of volatility, noise, and valence. Raw data are superimposed on all summary figures.

## Acknowledgements

We would like to thank James Gunnell for his help in collecting the data. This study was funded by a MRC Clinician Scientist Fellowship awarded to MB (MR/N008103/1). MB was supported by the Oxford Health NIHR Biomedical Research Centre. The views expressed are those of the authors and not necessarily those of the NHS, the NIHR, or the Department of Health.

## Additional information

### Competing interests

Michael Browning: Has received travel expenses from Lundbeck for attending conferences and consultancy fromJansen, CHDR and Novartis. The other author declares that no competing interests exist.

### Funding

| Funder | Grant reference number | Author |
|---|---|---|
| Medical Research Council | MR/N008103/1 | Michael Browning |

The funders had no role in study design, data collection and interpretation, or the decision to submit the work for publication.

### Author contributions

Erdem Pulcu, Conceptualization, Formal analysis, Investigation, Writing – review and editing; Michael Browning, Conceptualization, Software, Formal analysis, Supervision, Funding acquisition, Investigation, Visualization, Methodology, Writing – original draft

### Author ORCIDs

Erdem Pulcu ⓘ https://orcid.org/0000-0002-2170-0677
Michael Browning ⓘ https://orcid.org/0000-0001-9108-3144

### Ethics

The studywas approved by the University of Oxford Central Research Ethics Committee (R49753/RE001).

Reviewer #1 (Public review): https://doi.org/10.7554/eLife.103734.3.sa1
Reviewer #2 (Public review): https://doi.org/10.7554/eLife.103734.3.sa2
Author response https://doi.org/10.7554/eLife.103734.3.sa3

## Additional files

### Supplementary files

MDAR checklist

## Data availability

Study data and analysis scripts, including code for the various models used are available at: https://osf.io/j7md3/.

The following dataset was generated:

| Author(s) | Year | Dataset title | Dataset URL | Database and Identifier |
|---|---|---|---|---|
| Browning M | 2020 | Affective Variability | https://osf.io/j7md3/ | Open Science Framework, j7md3 |

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
