## [Editor Report · eLife Assessment]

This study makes an **important** contribution by showing that humans adapt learning rates rationally to environmental volatility yet systematically misattribute noise as volatility, demonstrating approximate rationality with simplified internal models. The evidence is **compelling**, encompassing a cleverly designed volatility-versus-noise paradigm, innovative lesion-based comparisons between reinforcement-learning and degraded Bayesian Observer Models, and convergent behavioural and pupillometric data. Expanding formal model comparisons (e.g., BIC/AIC) and directly contrasting RL and Bayesian fits to physiological markers would further enhance the work, but these are minor limitations that do not detract from the core findings.

---

## [Referee Report · Reviewer #1 (Public review)]

Summary:

The authors present an interesting study using RL and Bayesian modelling to examine differences in learning rate adaptation in conditions of high and low volatility and noise respectively. Through "lesioning" an optimal Bayesian model, they reveal that apparently suboptimal adaptation of learning rates results from incorrectly detecting volatility in the environment when it is not in fact present.

Strengths:

The experimental task used is cleverly designed and does a good job of manipulating both volatility and noise. The modelling approach takes an interesting and creative approach to understand the source of apparently suboptimal adaptation of learning rates to noise, through carefully "lesioning" and optimal Bayesian model to determine which components are responsible for this behaviour.

Weaknesses:

The model space could be more extensive, although the authors have covered the most relevant models for the question at hand.

Comments on revisions: I have no further recommendations for the authors, they have addressed my previous comments very well.

---

## [Referee Report · Reviewer #2 (Public review)]

Summary:

In this study, the authors aimed to investigate how humans learn and adapt their behavior in dynamic environments characterized by two distinct types of uncertainty: volatility (systematic changes in outcomes) and noise (random variability in outcomes). Specifically, they sought to understand how participants adjust their learning rates in response to changes in these forms of uncertainty.

To achieve this, the authors employed a two-step approach:

Reinforcement Learning (RL) Model:

They first used an RL model to fit participants' behavior, revealing that the learning rate was context-dependent-it varied based on the levels of volatility and noise. However, the RL model showed that participants misattributed noise as volatility, leading to higher learning rates in noisy conditions, where the optimal strategy would be to be less sensitive to random fluctuations.

Bayesian Observer Model (BOM):

To better account for this context dependency, they introduced a Bayesian Observer Model (BOM), which models how an ideal Bayesian learner would update their beliefs about environmental uncertainty. They found that a degraded version of the BOM, where the agent had a coarser representation of noise compared to volatility, best fit the participants' behavior. This suggested that participants were not fully distinguishing between noise and volatility, instead treating noise as volatility and adjusting their learning rates accordingly.

The authors also aimed to use pupillometry data (measuring pupil dilation) as a physiological marker to arbitrate between models and understand how participants' internal representations of uncertainty influenced both their behavior and physiological responses. Their objective was to explore whether the BOM could explain not just behavioral choices but also these physiological responses, thereby providing stronger evidence for the model's validity.

Overall, the study sought to reconcile approximate rationality in human learning by showing that participants still follow a Bayesian-like learning process, but with simplified internal models that lead to suboptimal decisions in noisy environments.

Strengths:

The generative model presented in the study is both innovative and insightful. The authors first employ a Reinforcement Learning (RL) model to fit participants' behavior, revealing that the learning rate is context-dependent-specifically, it varies based on the levels of volatility and noise in the task. They then introduce a Bayesian Observer Model (BOM) to account for this context dependency, ultimately finding that a degraded BOM-in which the agent has a coarser representation of noise compared to volatility-provides the best fit to the participants' behavior. This suggests that participants are not fully distinguishing between noise and volatility, leading to misattribution of noise as volatility. Consequently, participants adopt higher learning rates even in noisy contexts, where an optimal strategy would involve being less sensitive to new information (i.e., using lower learning rates). This finding highlights a rational but approximate learning process, as described in the paper.

Weaknesses:

While the RL and Bayesian models both successfully predict behavior, it remains unclear how to fully reconcile the two approaches. The RL model captures behavior in terms of a fixed or context-dependent learning rate, while the BOM provides a more nuanced account with dynamic updates based on volatility and noise. Both models can predict actions when fit appropriately, but the pupillometry data offers a promising avenue to arbitrate between the models. However, the current study does not provide a direct comparison between the RL framework and the Bayesian model in terms of how well they explain the pupillometry data. It would be valuable to see whether the RL model can also account for physiological markers of learning, such as pupil responses, or if the BOM offers a unique advantage in this regard. A comparison of the two models using pupillometry data could strengthen the argument for the BOM's superiority, as currently, the possibility that RL models could explain the physiological data remains unexplored.

The model comparison between the Bayesian Observer Model and the self-defined degraded internal model could be further enhanced. Since different assumptions about the internal model's structure lead to varying levels of model complexity, using a formal criterion such as Bayesian Information Criterion (BIC) or Akaike Information Criterion (AIC) would allow for a more rigorous comparison of model fit. Including such comparisons would ensure that the degraded BOM is not simply favored due to its flexibility or higher complexity, but rather because it genuinely captures the participants' behavioral and physiological data better than alternative models. This would also help address concerns about overfitting and provide a clearer justification for using the degraded BOM over other potential models.

Comments on revisions:

The authors have addressed all my questions. Congratulations on the impressive work accomplished by the authors!

---

## [Author Response]

The following is the authors’ response to the original reviews

**Public Reviews:**

**Reviewer #1 (Public review):**
Summary:The authors present an interesting study using RL and Bayesian modelling to examine differences in learning rate adaptation in conditions of high and low volatility and noise respectively. Through "lesioning" an optimal Bayesian model, they reveal that apparently a suboptimal adaptation of learning rates results from incorrectly detecting volatility in the environment when it is not in fact present.Strengths:The experimental task used is cleverly designed and does a good job of manipulating both volatility and noise. The modelling approach takes an interesting and creative approach to understanding the source of apparently suboptimal adaptation of learning rates to noise, through carefully "lesioning" and optimal Bayesian model to determine which components are responsible for this behaviour.We thank the reviewer for this assessment.Weaknesses:The study has a few substantial weaknesses; the data and modelling both appear robust and informative, and it tackles an interesting question. The model space could potentially have been expanded, particularly with regard to the inclusion of alternative strategies such as those that estimate latent states and adapt learning accordingly.

We thank the reviewer for this suggestion. We agree that it would be interesting to assess the ability of alternative models to reproduce the sub-optimal choices of participants in this study. The Bayesian Observer Model described in the paper is a form of Hierarchical Gaussian Filter, so we will assess the performance of a different class of models that are able to track uncertainty-- RL based models that are able to capture changes of uncertainty (the Kalman filter, and the model described by Cochran and Cisler, Plos Comp Biol 2019). We will assess the ability of the models to recapitulate the core behaviour of participants (in terms of learning rate adaption) and, if possible, assess their ability to account for the pupillometry response.

**Reviewer #2 (Public review):**
Summary:In this study, the authors aimed to investigate how humans learn and adapt their behavior in dynamic environments characterized by two distinct types of uncertainty: volatility (systematic changes in outcomes) and noise (random variability in outcomes). Specifically, they sought to understand how participants adjust their learning rates in response to changes in these forms of uncertainty.To achieve this, the authors employed a two-step approach:(1) Reinforcement Learning (RL) Model: They first used an RL model to fit participants' behavior, revealing that the learning rate was context-dependent. In other words, it varied based on the levels of volatility and noise. However, the RL model showed that participants misattributed noise as volatility, leading to higher learning rates in noisy conditions, where the optimal strategy would be to be less sensitive to random fluctuations.(2) Bayesian Observer Model (BOM): To better account for this context dependency, they introduced a Bayesian Observer Model (BOM), which models how an ideal Bayesian learner would update their beliefs about environmental uncertainty. They found that a degraded version of the BOM, where the agent had a coarser representation of noise compared to volatility, best fit the participants' behavior. This suggested that participants were not fully distinguishing between noise and volatility, instead treating noise as volatility and adjusting their learning rates accordingly.The authors also aimed to use pupillometry data (measuring pupil dilation) as a physiological marker to arbitrate between models and understand how participants' internal representations of uncertainty influenced both their behavior and physiological responses. Their objective was to explore whether the BOM could explain not just behavioral choices but also these physiological responses, thereby providing stronger evidence for the model's validity.Overall, the study sought to reconcile approximate rationality in human learning by showing that participants still follow a Bayesian-like learning process, but with simplified internal models that lead to suboptimal decisions in noisy environments.Strengths:The generative model presented in the study is both innovative and insightful. The authors first employ a Reinforcement Learning (RL) model to fit participants' behavior, revealing that the learning rate is context-dependent-specifically, it varies based on the levels of volatility and noise in the task. They then introduce a Bayesian Observer Model (BOM) to account for this context dependency, ultimately finding that a degraded BOM - in which the agent has a coarser representation of noise compared to volatility - provides the best fit for the participants' behavior. This suggests that participants do not fully distinguish between noise and volatility, leading to the misattribution of noise as volatility. Consequently, participants adopt higher learning rates even in noisy contexts, where an optimal strategy would involve being less sensitive to new information (i.e., using lower learning rates). This finding highlights a rational but approximate learning process, as described in the paper.

We thank the reviewer for their assessment of the paper.

Weaknesses:While the RL and Bayesian models both successfully predict behavior, it remains unclear how to fully reconcile the two approaches. The RL model captures behavior in terms of a fixed or context-dependent learning rate, while the BOM provides a more nuanced account with dynamic updates based on volatility and noise. Both models can predict actions when fit appropriately, but the pupillometry data offers a promising avenue to arbitrate between the models. However, the current study does not provide a direct comparison between the RL framework and the Bayesian model in terms of how well they explain the pupillometry data. It would be valuable to see whether the RL model can also account for physiological markers of learning, such as pupil responses, or if the BOM offers a unique advantage in this regard. A comparison of the two models using pupillometry data could strengthen the argument for the BOM's superiority, as currently, the possibility that RL models could explain the physiological data remains unexplored.

We thank the reviewer for this suggestion. In the current version of the paper, we use an extremely simple reinforcement learning model to simply measure the learning rate in each task block (as this is the key behavioural metric we are interested in). As the reviewer highlights, this simple model doesn’t estimate uncertainty or adapt to it. Given this, we don’t think we can directly compare this model to the Bayesian Observer Model—for example, in the current analysis of the pupillometry data we classify individual trials based on the BOM’s estimate of uncertainty and show that participants adapt their learning rate as expected to the reclassified trials, this analysis would not be possible with our current RL model. However, there are more complex RL based models that do estimate uncertainty (as discussed above in response to Reviewer #1) and so may more directly be compared to the BOM. We will attempt to apply these models to our task data and describe their ability to account for participant behaviour and physiological response as suggested by the Reviewer.

The model comparison between the Bayesian Observer Model and the self-defined degraded internal model could be further enhanced. Since different assumptions about the internal model's structure lead to varying levels of model complexity, using a formal criterion such as Bayesian Information Criterion (BIC) or Akaike Information Criterion (AIC) would allow for a more rigorous comparison of model fit. Including such comparisons would ensure that the degraded BOM is not simply favored due to its flexibility or higher complexity, but rather because it genuinely captures the participants' behavioral and physiological data better than alternative models. This would also help address concerns about overfitting and provide a clearer justification for using the degraded BOM over other potential models.

Thank you, we will add this.

**Recommendations for the authors:**

**Reviewer #1 (Recommendations for the authors):**
For clarity, the methods would benefit from further detail of task framing to participants. I.e. were there explicit instructions regarding volatility/task contingencies? Or were participants told nothing?

We have added in the following explanatory text to the methods section (page 20), clarifying the limited instructions provided to participants:

“Participants were informed that the task would be split into 6 blocks, that they had to learn which was the best option to choose, and that this option may change over time. They were not informed about the different forms of uncertainty we were investigating or of the underlying structure of the task (that uncertainty varied between blocks).”

In the results, it would be useful to report the general task behavior of participants to get a sense of how they performed across different parts of the task. Also, were participants excluded if they didn't show evidence of learning adaptation to volatility?

We have added the following text reporting overall performance to the results (page 6):

“Participants were able to learn the best option to choose in the task, selecting the most highly rewarded option on an average of 71% of trials (range 65% - 74%).”

And the following text to the methods, confirming that participants were not excluded if they didn’t respond to volatility/noise (the failure in this adaptation is the focus of the current study) (page 19):

“No exclusion criteria related to task performance were used.”

The results would benefit from a more intuitive explanation of what the lesioning is trying to recapitulate; this can get quite technical and the objective is not necessarily clear, especially for the less computationally-minded reader.

We have amended the relevant section of the results to clarify this point (page 9):

“Having shown that an optimal learner adjusts its learning rate to changes in volatility and noise as expected, we next sought to understand the relative noise insensitivity of participants. In these analyses we “lesion” the BOM, to reduce its performance in some way, and then assess whether doing so recapitulates the pattern of learning rate adaptation observed for participants (Fig 3e). In other words, we damage the model so it performs less well and then assess whether this damage makes the behaviour of the BOM (shown in Fig 3f) more closely resemble that seen in participants (Fig 3e).”

The modelling might be improved by the inclusion of another class of model. Specifically, models that adapt learning rates in response to the estimation of latent states underlying the current task outcomes would be very interesting to see. In a sense, these are also estimating volatility through changeability of latent states, and it would be interesting to explore whether the findings could also be explained by an incorrect assumption that the latent state has changed when outcomes are noisy.

Thank you for this suggestion. We have added additional sections to the supplementary materials in which we use a general latent state model and a simple RL model to try to recapitulate the behaviour of participants (and to compare with the BOM). These additional sections are extensive, so are not reproduced here. We have also added in a section to the discussion in the main paper covering this interesting question in which we confirm that we were unable to reproduce participant behaviour (or the normative effect of the lesioned BOMs) using these models but suggest that alternative latent state formulations would be interesting to explore in future work (page 18):

“A related question is whether other, non-Bayesian model formulations may be able to account for participants’ learning adaptation in response to volatility and noise. Of note, the reinforcement learning model used to measure learning rates in separate blocks does not achieve this goal—as this model is fitted separately to each block rather than adapting between blocks (NB the simple reinforcement learning model that is fitted across all blocks does not capture participant behaviour, see supplementary information). One candidate class of model that has potential here is latent-state models (Cochran & Cisler, 2019), in which the variance and unexpected changes in the process being learned (which have a degree of similarity with noise and volatility respectively) is estimated and used to alter the model’s rates of updating as well as the estimated number of states being considered. Using the model described by Cochran and Cisler, we were unable to replicate the learning rate adaptation demonstrated by participants in the current study (see supplementary information) although it remains possible that other latent state formulations may be more successful.”

The discussion may benefit from a little more discussion of where this work leads us - what is the next step?

As above, we have added in a suggestion about future modelling work. We have also added in a section about the outstanding interesting questions concerning the neural representation of these quantities, reproduced in response to the suggestion by reviewer #2 below.

**Reviewer #2 (Recommendations for the authors):**
The study presents an opportunity to explore potential neural coding models that could account for the cognitive processes underlying the task. In the field of neural coding, noise correlation is often measured to understand how a population of neurons responds to the same stimulus, which could be related to the noise signal in this task. Since the brain likely treats the stimulus as the same, with noise representing minor changes, this aspect could be linked to the participants' difficulty distinguishing noise from volatility. On the other hand, signal correlation is used to understand how neurons respond to different stimuli, which can be mapped to the volatility signal in the task. It would be highly beneficial if the authors could discuss how these established concepts from neural population coding might relate to the Bayesian behavior model used in the study. For instance, how might neurons encode the distinction between noise and volatility at a population level? Could noise correlation lead to the misattribution of noise as volatility at a neural level, mirroring the behavioral findings? Discussing possible neural models that could explain the observed behavior and relating it to the existing literature on neural population coding would significantly enrich the discussion. It would also open up avenues for future research, linking these behavioral findings to potential neural mechanisms.

We thank the reviewer for this interesting suggestion. We have added in the following paragraph to the discussion section which we hope does justice to this interesting questions (page 18):

“Previous work examining the neural representations of uncertainty have tended to report correlations between brain activity and some task-based estimate of one form of uncertainty at a time (Behrens et al., 2007; Walker et al., 2020, 2023). We are not aware of work that has, for example, systematically varied volatility and noise and reported distinct correlations for each. An interesting possibility as to how different forms of uncertainty may be encoded is suggested by parallels with the neuronal decoding literature. One question addressed by this literature is how the brain decodes changes in the world from the distributed, noisy neural responses to those changes, with a particular focus on the influence of different forms of between-neuron correlation (Averbeck et al., 2006; Kohn et al., 2016). Specifically, signal-correlation, the degree to which different neurons represent similar external quantities (required to track volatility) is distinguished from, and often limited by, noise-correlation, the degree to which the activity of different neurons covaries independently of these external quantities. One possibility relevant to the current study, which resembles the underlying logic of the BOM, is that a population of neurons represents the estimated mean of the generative process that produces task outcomes. In this case, volatility would be tracked as the signal-correlation across this population, whereas noise would be analogous to the noise-correlation and, crucially, misestimation of noise as volatility might arise as misestimation of these two forms of correlation. While the current study clearly cannot adjudicate on the neural representation of these processes, our finding of distinct behavioural and physiological responses to the two forms of uncertainty, does suggest that separable neural representations of uncertainty are maintained.”